# Exosomal Dynamics and Brain Redox Imbalance: Implications in Alzheimer’s Disease Pathology and Diagnosis

**DOI:** 10.3390/antiox13030316

**Published:** 2024-03-05

**Authors:** Aritri Bir, Arindam Ghosh, Aman Chauhan, Sarama Saha, Adesh K. Saini, Marco Bisaglia, Sasanka Chakrabarti

**Affiliations:** 1Department of Biochemistry, Dr B. C. Roy Multi-Speciality Medical Research Centre, Indian Institute of Technology Kharagpur, Kharagpur 721302, West Bengal, India; aritri@bcrmrc.iitkgp.ac.in (A.B.); arindam@bcrmrc.iitkgp.ac.in (A.G.); 2Department of Biochemistry and Central Research Cell, Maharishi Markandeshwar Institute of Medical Sciences and Research, Maharishi Markandeshwar (Deemed to be University), Mullana-Ambala 133207, Haryana, India; dr.amanchauhan@mmumullana.org; 3Department of Biochemistry, All India Institute of Medical Science (AIIMS), Rishikesh 174001, Uttarakhand, India; arama.bchem@aiimsrishikesh.edu.in; 4Department of Biotechnology, Maharishi Markandeshwar (Deemed to be University), Mullana-Ambala 133207, Haryana, India; sainiade@mmumullana.org; 5Department of Biology, University of Padova, Via Ugo Bassi 58/B, 35131 Padova, Italy; 6Study Center for Neurodegeneration (CESNE), 35121 Padova, Italy

**Keywords:** Alzheimer’s disease, amyloid beta, exosomes, miRNA, mitochondria, oxidative stress, tau phosphorylation

## Abstract

Oxidative burden plays a central role in Alzheimer’s disease (AD) pathology, fostering protein aggregation, inflammation, mitochondrial impairment, and cellular dysfunction that collectively lead to neuronal injury. The role of exosomes in propagating the pathology of neurodegenerative diseases including AD is now well established. However, recent studies have also shown that exosomes are crucial responders to oxidative stress in different tissues. Thus, this offers new insights and mechanistic links within the complex pathogenesis of AD through the involvement of oxidative stress and exosomes. Several studies have indicated that exosomes, acting as intracellular communicators, disseminate oxidatively modified contents from one cell to another, propagating the pathology of AD. Another emerging aspect is the exosome-mediated inhibition of ferroptosis in multiple tissues under different conditions which may have a role in neurodegenerative diseases as well. Apart from their involvement in the pathogenesis of AD, exosomes enter the bloodstream serving as novel noninvasive biomarkers for AD; some of the exosome contents also reflect the cerebral oxidative stress in this disease condition. This review highlights the intricate interplay between oxidative stress and exosome dynamics and underscores the potential of exosomes as a novel tool in AD diagnosis.

## 1. Introduction

Alzheimer’s disease is the most prevalent form of dementia and a major neurodegenerative disorder. It presently impacts an estimated 50 million individuals globally. However, projections indicate that the incidence will triple by 2050 as a result of the population’s natural aging process, with low- and middle-income countries experiencing the most substantial surge in incidence [1]. In addition, Alzheimer’s patients require costly and specialized treatment; the global annual cost of treatment approaches one trillion US dollars, and this figure is expected to increase substantially by 2030 [2]. Over the last few decades, research has indicated that the pathogenesis of AD is influenced by a variety of factors, which include biological elements (e.g., aging, gender and body weight), environmental components (e.g., lifestyle, toxins and brain injury), and genetic components (e.g., susceptibility genetic polymorphisms in sporadic cases and Amyloid beta precursor protein, presenilin 1, and presenilin 2 genetic mutations in familial AD) [3]. Although our comprehension of AD has been significantly expanded by the accumulation of knowledge, the underlying mechanism of AD pathogenesis remains confusing. Recent research has shed light on the pivotal role played by oxidative stress in AD pathology, acting as a catalyst for protein aggregation, inflammation, mitochondrial dysfunction, and overall cellular impairment that collectively culminate in neuronal injury [4,5,6,7]. Workgroups convened by the Alzheimer’s Association and the National Institute on Aging produced diagnostic guidelines for the entire disease continuum in 2011 [8]. Additionally, in 2018, these groups developed a research framework to advance the hypothesis of AD as a biological disease [9]. According to the Research Framework, AD is characterized by its fundamental pathological processes, which may be seen with postmortem examination or through biomarkers while the person is still alive. Biomarkers are categorized according to their association with amyloid-beta (Aβ) deposition, pathologic Tau, and neurodegeneration (ATN). The ATN classification system categorizes many biomarkers, including imaging and biofluids, based on the specific pathological process that each biomarker assesses [9].

The utilization of current biomarkers often involves invasive procedures such as the collection of cerebrospinal fluid (CSF) via a lumbar puncture or costly imaging techniques. The imaging diagnostic modalities of AD, such as positron emission tomography (PET) scan for amyloid deposition in the brain or cerebral ^18^F-2-deoxyglucose (FDG) uptake via an FDG-PET scan, are expensive and not easily available [10]. Further, the existing biomarkers may also fail to detect AD at an early stage before significant brain damage occurs, which is crucial for the effective management of the disease and the identification of neuroprotective drugs through clinical trials. Despite extensive research and costly clinical trials, amyloid-targeted therapeutics have generally failed and many investigational programs were abandoned in the last decade. The latest drug lecanemab (accelerated approval from the FDA) has also been shrouded in criticisms [11]. No effective disease-modifying treatment for AD has emerged, and this may have failed due to late initiation, incorrect target selection, and a lack of understanding of AD’s complex pathophysiology [12]. Thus, newer avenues are to be searched which would lead to a better understanding of the pathogenesis of AD and the identification of novel and simple early diagnostic markers.

Once considered to be nothing more than cellular debris, exosomes have recently been recognized to play a significant role in the process of molecular communication both within and between cells, playing an important role in the pathogenesis of multiple diseases [13]. The involvement of exosomes in the progression of disease pathology within the central nervous system (CNS) in several neurodegenerative diseases including AD has also been well established [14,15]. On the other hand, fascinating observations have been made on exosomes in relation to oxidative stress. For instance, mesenchymal stem-cell-derived exosomes have enhanced the antioxidant capacities and alleviated cellular damage induced by oxidative stress [16]. There are other interactions of oxidative stress and exosomes which we will discuss later in this review. Importantly, this offers fresh perspectives and mechanistic connections within the complex pathogenesis of AD for the first time and also indicates the possibility of detecting simpler exosome-based AD biomarkers in the peripheral circulation. The current review is intended to delve into these aspects.

## 2. Oxidative Stress and Alzheimer’s Disease

Oxidative stress is a condition when the balance between the production of reactive oxygen species (ROS) and their detoxification via the antioxidant defense system is lost in favor of increased levels of ROS in the tissue. Under normal conditions of redox balance, ROS are inactivated nearly completely by the cellular antioxidants leaving presumably only a small amount of these reactive oxy-radicals capable of taking part in redox signaling processes involved in cell growth, proliferation, differentiation and death [17,18]. The major sources of intracellular ROS are the mitochondrial electron transport chain, the NADPH oxidase (NOX) complex having membrane-bound and cytosolic components, the cyclooxygenase and lipoxygenase catalyzed reactions, cytochrome P450 dependent reactions, xanthine oxidase reaction and peroxisomal fatty acid oxidation [17,19]. Conventionally, the superoxide radicals (O_2_^−^), H_2_O_2_, hydroxyl radicals (OH^.^) and singlet oxygen are considered members of ROS, but the term has been expanded to include many other reactive molecules or free radicals such as lipid-derived radicals, lipid and protein hydroperoxides, peroxynitrite and many others [20]. The inter-conversions of different ROS and their complex reactions with various biomolecules like phospholipids, proteins and nucleic acids are catalyzed at multiple steps by transition metals. For example, Fe^2+^ can catalyze the decomposition of H_2_O_2_ via Fenton’s reactions or catalyze the formation of alkoxyl and peroxyl radicals during lipid peroxidation chain reaction [21,22]. On the other hand, the final inactivation of ROS occurs via different cellular antioxidants consisting of an array of enzymes (superoxide dismutase, catalase, glutathione peroxidase, etc.) or proteins (peroxiredoxins and thioredoxin) or nonprotein antioxidants (α-tocopherol, retinol, ascorbic acid, bilirubin, melatonin and others) present in the tissue [21,22]. During oxidative stress, as a result of redox imbalance, the excess ROS can cause direct oxidative damage to cellular components (membranes, enzymes, ion channels, etc.), initiate aberrant redox signaling, trigger a more regulated cell death pathway, such as ferroptosis, or aggravate an inflammatory reaction [18,23,24,25,26]. This complex scenario is probably a part of the pathogenesis of different diseases including neurodegenerative disorders.

The involvement of oxidative stress and redox imbalance in the pathogenesis of AD is highlighted by the multiple lines of compelling evidence. For example, the evidence of extensive oxidative damage was noticed in postmortem brains of individuals with AD with a substantial buildup of oxidative damage markers of phospholipids, proteins and nucleic acids, such as malondialdehyde, 4-hydroxynonenal, F_2_-isoprostane, protein carbonyls, nitro-tyrosine, 8-hydroxydeoxyguanosine and others, respectively [27,28,29,30,31,32]. Additionally, an elevation in the levels of transition metals, such as iron (Fe), has been documented in the AD brain and some reports have indicated a deficiency of antioxidant enzymes especially glutathione peroxidase and catalase [6,33,34,35,36,37]. The utilization of redox proteomics in the analysis of postmortem AD brains further provided additional evidence of oxidative damage to multiple enzymes and proteins implicated in energy metabolism, neurotransmission, mitochondrial and synaptic functions and proteasomal functions [38,39]. In transgenic AD mice, such evidence of oxidative damage has been reported along with the deposition of Aβ [40,41]. However, it must be pointed out that some meta-analysis-based publications have questioned the oxidative damage hypothesis of AD [42,43].

### 2.1. Accumulation of Amyloid Beta Due to Oxidative Stress

The reciprocal impact of Aβ and oxidative stress on each other has played a significant role in the development of AD. The interaction between oxidative stress and Aβ proteinopathy in AD is complex, impacting several phases of Aβ production and processing. This encompasses the control of gene expression for the Aβ precursor protein (APP) gene, the conversion of APP mRNA into protein and the subsequent breakdown and elimination of both APP and Aβ peptides. Experimental studies have shown that transcription factors that respond to ROS, such as heat shock factor-1 (HSF-1) and nuclear factor-kappa B (NF-kB), are involved in stimulating the expression of the APP gene by attaching to the promoter regions of the gene. ROS can also participate in the modulation of the expression of both β-secretase (BACE1) and γ-secretase. For instance, oxidative stress enhances BACE1 activity by affecting the process of protein synthesis, specifically through the activation of double-stranded RNA-dependent protein kinase (PKR) and eukaryotic initiation factor-2 (eIF2) phosphorylation [44]. Accordingly, increased BACE1 activity has been detected in the presence of oxidative-stress-inducing substances, such as 4-hydroxynonenal (4-HNE), hydrogen peroxide (H_2_O_2_) and iron [45]. The γ-secretase enzyme complex, which plays a crucial role in the release of Aβ42 from APP, is similarly affected by oxidative stress. Specifically, presenilin 1 (PS1) levels, a component of the γ-secretase complex, increase in the presence of oxidative stress conditions [46]. The production of APP is also regulated at the post-transcriptional level, through the presence of an iron-responsive element (IRE) located in the 5’-untranslated region (UTR) of APP mRNA. The IRE-binding protein (IREBP) controls the process of translation by inhibiting it through the binding with the IRE sequence. However, when the levels of iron inside the cell become high, the IREBP detaches from the binding site, resulting in an increased translation. This connection between oxidative stress, iron levels, and APP production in the brains of individuals with AD has been well-established and characterized [47].

In addition to the production of Aβ, oxidative stress also affects the removal of Aβ from the brain, by acting on specific receptors, such as the low-density lipoprotein receptor-related protein 1 (LRP1) and the receptor for advanced glycation end products (RAGE), which play a key role in mediating the transport of Aβ across the blood–brain barrier, participating in its clearance [48]. In this frame, changes in the membrane-bound form of LRP1 have been shown to impede the effective removal of Aβ from the brain. In addition, the oxidation of the soluble form of LRP1 in the blood, which binds Aβ and acts as a sort of sink, loses its ability to efficiently bind to circulating Aβ, which can enter the brain again [48].

To summarize, the complex connection between oxidative stress and different aspects of Aβ synthesis, processing and removal provides insight into the complicated processes involved in the development of AD. These results greatly enhance our comprehension of the many factors that contribute to neurodegeneration in AD caused by oxidative stress.

### 2.2. Induction of Oxidative Stress Mediated by Amyloid Beta

Numerous mechanisms have been described which contribute to the initiation of oxidative damage in the AD brain, with a prominent role attributed to Aβ-induced ROS generation, as extensively documented in various experimental studies [6,49,50,51,52,53]. Both Aβ42 and Aβ40 possess the ability to bind transition metals in a redox-active form through specific amino acid residues, including His6, His13, and His14. The resulting coordination chemistry facilitates redox-cycling reactions that generate ROS, a process observed in experimental settings [54,55]. This is particularly noteworthy given the postmortem evidence of elevated levels of redox-active transition metals, such as Fe in the AD brain, especially in proximity to plaques [55]. However, it is worth mentioning that an alternative perspective was earlier published where the possible antioxidative and protective functions of Aβ were investigated [56]. This hypothesis is supported by scattered experimental evidence showing an antioxidant and protective role of Aβ, which has been shown to scavenge reactive radicals of lipid oxidation, prevent ROS formation by sequestering transition metals or even block mitochondrial oxygen-free radical production [57]. Another study has shown the pro-oxidative nature of Aβ oligomers and the antioxidative properties of monomers and fibrils [58]. Various other in vitro studies that indicate an antioxidant and protective role of Aβ have been summarized elsewhere [59].

Beyond the direct production of ROS by bound redox-active metals, Aβ may induce intracellular ROS production leading to neuronal death through the involvement of Apoptosis signal-regulating kinase 1 (ASK1) [60]. Moreover, experimental studies indicate that Aβ increases ROS production, potentially through the activation of NOX and enhanced mitochondrial production of oxygen radicals. Notably, mitochondria-targeted antioxidants can prevent this effect [61]. Other mechanisms of Aβ-induced ROS production involve the stimulation of microglial cells by soluble or fibrillar forms of Aβ [62]. In primary microglial cultures or cocultures with neurons, activated microglia can generate ROS and proinflammatory cytokines such as interleukin-6 (IL-6), IL-1β, and tumor necrosis factor-alpha (TNF-α), thus triggering an inflammatory reaction [63]. In addition, fibrillar Aβ has been described to interact and activate the scavenger receptor CD36 on microglial cells, causing an increase in the formation of ROS, the release of cytokines and phagocytosis by microglial cells [64]. Furthermore, recent research has shown that the macrophage antigen-1 (MAC-1) receptor and Phosphoinositide 3-kinases (PI3K) are involved in the process of Aβ-induced microglial activation and ROS generation, leading to the activation of NOX. The formation of ROS by microglia in response NOX oxidase [63,65].

### 2.3. Role of Tau Phosphorylation in Oxidative Stress

One characteristic of the development of AD is the creation of neurofibrillary tangles that occurs when there is an excessive buildup of hyperphosphorylated Tau protein within the cells [66]. This process plays a substantial role in the deterioration of nerve fibers and impaired communication between neurons, one of the pathological features of AD. The protein Tau, which is linked with microtubules, possesses many potential phosphorylation sites in its C-terminal and proline-rich regions [67]. Several kinases, namely Glycogen synthase kinase 3β (GSK3β) and Cyclin Dependent Kinase 5 (CDK5), can phosphorylate Tau, while phosphatases, such as Protein phosphatase 2A (PP2A), PP1 and PP2B, may remove phosphate groups from it. Elevated activities of GSK3β and CDK5, together with diminished PP2A activity, seem to be responsible for the increased Tau phosphorylation seen in AD [68,69]. Nevertheless, the factors behind these changes in kinase and phosphatase activity in AD are still not well understood.

Experimental models have investigated the influence of oxidative stress on the process of Tau phosphorylation. In this frame, glutathione depletion has been shown to increase Tau phosphorylation in cultured M17 neuroblastoma cells [70,71], while the process may be alleviated by the antioxidant trolox [45]. Accordingly, the phosphorylation of Tau appears to increase in rat primary cortical neuronal culture when exposed to a mixture of Fe^2+^ and H_2_O_2_ [67,72], but contradictory data also exist, suggesting that Tau phosphorylation decreases in many experimental models when exposed to oxidative stress [73]. In conclusion, the precise connection between oxidative stress and Tau phosphorylation remains somewhat ambiguous, with the involvement of kinases (such as GSK3 or CDK5) or phosphatases (such as PP1) influencing this intricate association.

### 2.4. Mitochondria Dysfunction, Oxidative Stress and AD

Mitochondrial dysfunction stands as a pivotal mechanism in AD pathogenesis, validated with studies involving postmortem AD brains and various experimental models [74,75]. Structural alterations in mitochondria, such as fragmentation with abnormal cristae, compromised bioenergetics, reduced enzyme activities, impaired ATP synthesis, mitochondrial membrane depolarization, increased ROS production and disturbed mitochondrial biogenesis and dynamics have been observed [45]. Given that mitochondrial oxidative phosphorylation is a major source of ROS, it is reasonable to infer that mitochondrial dysfunction significantly contributes to oxidative stress in the AD brain [76,77].

The interrelation between mitochondrial dysfunction and proteotoxicity, mainly involving Aβ, represents a subject of intense research. In transgenic AD models, progressive Aβ accumulation in brain mitochondria correlates with diminished respiratory chain enzyme activities and decreased oxygen consumption rates [75]. Aβ has been shown to bind to mitochondrial short-chain alcohol dehydrogenase, known as Aβ-binding alcohol dehydrogenase (ABAD), as well as heat shock protein 60 (Hsp60), in the mitochondrial matrix [78]. The accumulation of Aβ may also inhibit the mitochondrial peptidasome (PreP), impacting the processing of mitochondrially targeted protein presequences and leading to multiple functional mitochondrial anomalies [79]. In addition, soluble Aβ oligomers can impair mitochondrial functions, possibly due to interactions with various mitochondrial proteins, including adenine nucleotide translocase, components of the translocase of the outer membrane (TOM) as well as inner membrane (TIM), and cyclophilin D [80]. Aβ oligomers may also create membrane-spanning channels (amyloid pores), contributing to toxic effects on mitochondria [81]. Another potential pathological mechanism could be related to the blockage of mitochondrial protein import channels by APP, hindering the entry of nuclear DNA-coded proteins, including respiratory chain complex subunits [82]. In astrocytes, Aβ-induced mitochondrial dysfunction seems to involve cytosolic and calcium-independent phospholipase A2, while the activation of NOX by Aβ has been described to enhance ROS production, potentially causing mitochondrial dysfunction and glutathione depletion in both neurons and astrocytes [83,84]. These findings underscore the interesting relationship between mitochondrial dysfunction and Aβ-induced pathology in AD.

### 2.5. Metal Ion Homeostasis and Oxidative Stress in AD

Transition metals, comprising copper (Cu) and iron (Fe), are indispensable for diverse biological functions of brain activity including synapse regulation, myelination, synaptogenesis and synaptic plasticity [27]. Predominantly, Fe and Cu remain bound to cellular proteins, but a very minute amount remains in the labile or redox-active form to take part in various ROS-mediated reactions, including Fenton’s reaction, to generate highly reactive OH· radicals from H_2_O_2_. Thus, the delicate homeostasis of redox-active metal is crucial due to this potential hazard of toxic free radicals instigating oxidative stress and consequent alterations harmful to neuronal cells [85,86,87].

In the context of AD, perturbations in Fe homeostasis in the brain have already been mentioned earlier. Such changes in Fe level could be region-specific in the AD brain and may correlate with Aβ deposition, Tau accumulation or neurodegeneration [34,35,88]. Many other studies have established at the microscopic level that Fe was deposited within amyloid plaques in AD patients and transgenic mouse models, intimating a link between AD pathology and the interactions of Aβ with Fe [55,89,90]. Unlike Fe, multiple studies, however, failed to establish a clearly elevated level of Cu in the postmortem AD brain [55,91,92]. Nevertheless, it is important to note that both Cu^2+^ and Fe^3+^ may bind to Aβ in a redox-active form and can generate ROS [54,55]. Initially, the liganded metal is reduced presumably by a methionine residue of Aβ or with the help of an endogenous reducing compound; the reduced metal ion may then be oxidized via a reaction with molecular oxygen generating O_2_^−^ or with H_2_O_2_ producing OH· radicals through Fenton’s reaction [55]. Thus, a redox cycling of the Aβ liganded Fe^3+^ or Cu^2+^ could be an important source of ROS in the AD brain. Importantly, the binding of Cu^2+^ to Aβ monomers induces structural changes, amplifying peptide aggregation. This aberrant interaction may also generate ROS, contributing to oxidative stress in Aβ-mediated neurotoxicity [55]. Furthermore, Fe/Cu homeostasis intertwines with APP generation and processing, influencing the accumulation and the release of Aβ and affecting the production of free radicals [93,94].

Alterations at the levels of metal transporters might also exacerbate abnormal metal homeostasis in AD. For instance, increased levels of the divalent metal transporter 1 (DMT1) in AD brains and APP transgenic mice have been shown to correlate with elevated intracellular Fe levels, oxidative stress and cellular toxicity [95]. Overall, an elevated Fe level in the AD brain may induce oxidative damage, initiate the process of ferroptotic death of neurons and also affect the accumulation of Aβ and Tau.

### 2.6. Abnormal Glucose Metabolism, Oxidative Stress and AD

Recent studies have delved into the intricacies of glucose metabolism impairment in AD and amnestic Mild Cognitive Impairment (MCI) brains, pinpointing the close association between inefficient glucose utilization, oxidative damage and diminished ATP production [96]. Oxidative modifications to crucial enzymes in glycolysis and the tricarboxylic acid (TCA) cycle, as well as mitochondrial dysfunction, collectively contribute to the overall decrease in energy production. The identified oxidative modifications in the AD brain include those to glycolytic enzymes like aldolase, triosephosphate isomerase and glyceraldehyde-3-phosphate dehydrogenase, along with TCA cycle enzyme aconitase, creatine kinase and ATP synthase in brain mitochondria [97]. Given the high energy demand of the brain to sustain neuronal activity, a possible consequence of this decreased ATP production could hinder the maintenance of ionic gradients and impede the production and propagation of action potentials strongly affecting neuronal function [97]. The ensuing synaptic dysfunction and eventual neuronal death can be further exacerbated by the entry of extracellular Ca^2+^, creating ROS and leading to oxidative damage. Excessive Ca^2+^ levels might also trigger apoptosis, synaptic dysfunction and cognitive decline [98,99]. Some interesting studies in cell-based models of AD-like neurodegeneration induced glucose hypometabolic stress by inhibiting glycolysis by glyceraldehyde or cellular glucose uptake using an inhibitor of glucose transporters (WZB117); under such conditions, the increased ROS formation, mitochondrial dysfunctions, Ca^2+^ dysregulation, increased β-secretase expression and enhanced Aβ production could be noticed with eventual neural cell death [100,101]. Studies have also explored the role of mTOR activation in AD, linking it to insulin resistance, impaired autophagy, oxidative damage and neuronal death [102]. Finally, glycation processes and the formation of advanced glycation end products (AGEs) have been proven relevant to AD pathology, oxidative stress and vascular dysfunction [103].

## 3. Exosome Dynamics and Functional Significance

### 3.1. Biogenesis, Release and Transport of Exosomes

Exosomes are small extracellular vesicles that are secreted by various cells in the body, including immune cells, stem cells and cancer cells. They are approximately 100 nm (average) in diameter and are composed of a lipid bilayer that encapsulates a variety of biomolecules, including proteins, lipids and nucleic acids [104]. Exosomes have emerged as an important area of research in recent years, due to their diverse functions in intercellular communication and potential use as biomarkers for various diseases [105]. Exosomes are formed through the endosomal pathway, which begins with the invagination of the plasma membrane to form early endosomes. These early endosomes then mature into late endosomes, which contain intraluminal vesicles (ILVs) that are formed through the inward budding of the endosomal membrane. The ILVs can then be released from the late endosome as exosomes, through the fusion of the late endosome with the plasma membrane [106]. The composition of exosomes can vary depending on the cell of origin and the cellular and environmental conditions. However, exosomes typically contain a variety of proteins, including tetraspanins, heat shock proteins and endosomal sorting complexes required for transport (ESCRT) proteins [107]. Exosomes also contain lipids, including cholesterol, sphingomyelin and ceramide, which are important for the stability and function of the exosome membrane. Additionally, exosomes contain various nucleic acids, including microRNAs (miRNAs), mRNAs and other noncoding RNAs [108].

There is substantial evidence indicating that exosomes are largely transported over the blood–brain barrier via transcytosis, a process similar to how immune cells and infectious pathogens are transported [109]. Exosomes traverse the intracellular compartment via transcytosis, unlike paracellular routes that brain cells use to cross the extracellular region. Studies suggest two potential ways exosomes might enter the brain: either by passing completely past the endothelial cell barrier or by being trapped within the brain endothelial cells. Exosomes influence the whole brain, whereas sequestration impacts brain endothelial cells, leading to controlled and particular transport processes [109,110,111]. Exosomes have been shown to play diverse roles in intercellular communication and cellular signaling. One major function of exosomes is the transfer of molecules between cells, which can influence cell growth, differentiation and survival [104]. For example, exosomes secreted by stem cells have been shown to promote tissue repair and regeneration by delivering growth factors and other bioactive molecules to damaged cells. Exosomes are also involved in immune regulation and response [112,113]. Immune cells can secrete exosomes that contain cytokines, chemokines and other signaling molecules that can stimulate or suppress immune responses [114]. Additionally, exosomes can act as carriers for antigen presentation, which can activate immune responses against infectious agents or cancer cells [115].

Recent studies have also suggested that exosomes may play a role in neurodegenerative diseases, including AD [116]. Neuronal exosomes have been shown to contain neurotransmitters, neuromodulators and synaptic proteins, which may play a role in synaptic plasticity and communication between neurons [117,118]. Exosomes secreted by neurons and glial cells have also been shown to contain Aβ peptides and Tau protein [119,120] and may contribute to the spread of AD pathology throughout the brain.

### 3.2. Oxidative Stress Induces Changes in Exosomes

Exosomes act as dynamic responders to oxidative challenges by contributing both to cellular defense mechanisms and the propagation of oxidative damage. ROS imbalance modifies the process of biogenesis and the release of exosomes. Prolonged or severe oxidative stress can also alter their composition. Exosomes released under oxidative stress carry bioactive molecules that may influence the modulation of stressed cells and neighboring cells [121,122,123,124]. This bidirectional communication is crucial for maintaining cellular homeostasis under conditions of oxidative challenge.

Changes in redox status prompt an elevation in the amount of exosomes, achieved through either augmented secretion or diminished degradation. In fact, enhanced autophagy conditions steer multivesicular bodies (MVBs) towards lysosomes instead of the plasma membrane, hindering the release of exosomes [125,126], while the inhibition of autophagic trafficking promotes exosome release [127]. Accordingly, enhanced secretion of exosomes has been shown under oxidative stress in different cultured cell lines [128,129,130,131]. Moreover, oxidative stress induced by nanoparticles, mechanical injury or chemicals like tBHP, has been described to increase the amount of exosomes, which is sometimes associated with morphological changes as well [131]. The impact of oxidative stress on exosomes biogenesis and release is summarized in Figure 1.

Alterations in the intracellular redox state can also modulate exosome release with alterations in exosomal content. For instance, endogenous ROS induced by homocysteine has been shown to enhance the release of exosomes containing inflammatory cytokines [132].

Moreover, exosomes isolated from oxidatively challenged cells showed decreased content of prosurvival proteins and upregulated proapoptotic proteins, consistent with reported oxidative-stress-mediated alterations in the phosphorylation status of proteins governing cell proliferation, survival and energy metabolism [133]. Finally, ROS imbalance has been described as altering the lipid composition of exosomes, particularly the levels of oxidized lipids, and their propagation from cell to cell. Through the expression or transport of oxidatively modified lipids, exosomes were shown to impart both proinflammatory and anti-inflammatory effects over neighboring cells [134].

## 4. Exosomes in AD

Recent advances in the isolation and characterization of brain-derived exosomes (BDEs) from AD samples have revealed a significant potential of the exosomal cargo in bridging the knowledge gap between peripheral biomarkers and CNS pathology in AD. This suggests that BDEs have the potential to be used as diagnostic and prognostic biomarkers for AD.

### 4.1. Proteins of Exosomal Cargo in AD

The quality and quantity of CNS-specific exosomes and their contents are highly correlated with the progression of AD. Thus, they may aid in the development of more precise and early diagnosis of AD. In fact, the brain-derived exosomal cargo could aid in the diagnosis of AD even before the emergence of cognitive losses [135]. In this frame, it has been shown that exosome-bound Aβ levels could exhibit a stronger correlation with PET imaging of brain amyloid plaques and can be used to discriminate between distinct clinical stages of dementia more effectively than unbound or total circulating Aβ [136]. Similarly, the levels of Aβ1–42, total Tau and phosphorylated Tau at S181 (p-S181-Tau) or S396 (p-S396-Tau) in exosomes could differentiate patients with AD from those with MCI and/or controls [137,138]. Importantly, proteins like p-S181-Tau and p-S396-Tau in BDEs have shown greater efficiency than that of plasma-derived exosomes in distinguishing AD from controls [138,139].

Aβ peptide is made from APP via sequential proteolytic cleavages and a variety of post-translational modifications. Numerous reports have demonstrated the presence of full-length APP and its derivatives, including APP C-terminal fragments and the APP intracellular domain, within exosomes of different in vivo and in vitro AD models and in exosomes derived from neurons of AD patients [140,141,142]. Enzymes like BACE1, PS1 and PS2, relevant in amyloid processing, were also reported to be present in exosomes [15]. Following Aβ trafficking to MVBs, a small fraction of Aβ peptides is secreted from the cells in conjunction with exosomes. Interestingly, BACE1 was identified in released exosomes as well as its colocalization with early exosome markers in transgenic neuronal cell lines [143]. Notably, the presence of the exosome marker protein Alix in plaques of post-mortem AD brains provides support to the idea that the release of exosomes carrying Aβ might play a role in plaque development and the advancement of the disease [144]. Consistent with this notion, another study demonstrated the cytotoxic properties of exosomes isolated from postmortem AD brains [145]. The Aβ oligomer-containing exosomes were uptaken by neurons and then exosomal content was further propagated and released to neighboring cells [145].

The propagation of intracellular toxic Tau proteins to neighboring cells through extracellular seeds is considered an important mechanism for spreading AD pathology in different brain regions. Exosomes could play an important role in disseminating Tau-mediated neuronal damage. Pathological Tau seeds transported by exosomes have been shown in different transgenic mice models to cause misfolding and aggregation of monomeric Tau in recipient cells [146,147]. Similarly, the propagation of Tau inclusions and massive neurodegenerations in wild-type mouse brains have been observed following the injection of exosome suspensions derived from neuronally differentiated, human-derived induced pluripotent stem cells enriched in Tau P301L and V337M mutations [148]. Microglia seem to play a crucial role in the exosome-mediated transmission of Tau pathology by internalizing and secreting Tau-containing exosomes [149]. Microglial exosome production can also be influenced by genes associated with AD which can lead to the dissemination of exosomal Tau [150,151]. Studies have shown that Aβ42, t-tau, p-tau181 and other proteins isolated from peripheral exosomes derived from neurons may differentiate between people with AD or MCI and healthy persons [152,153,154]. The neuronal-derived peripheral exosomes were reported to contain significantly lower levels of synaptic proteins like synaptophysin, synaptotagmin, growth-associated protein 43 and synapsin-1 in AD than other neurodegenerative diseases like frontotemporal dementia [155]. Furthermore, altered levels of proteins related to lysosomes and autophagy in neuronally derived exosomes in peripheral circulation were demonstrated in preclinical AD in a longitudinal study [156]. It is tempting to speculate in this context, and given the bidirectional movement of exosomes, that peripheral exosomes carrying entrapped molecules such as proinflammatory cytokines or metabolites from gut microbiota or other bioactive molecules may enter the brain during the preclinical course of AD and may play a role in inducing neuroinflammation or neurodegenerative process in the CNS. The role played by exosomes in propagating Aβ and Tau pathology is summarized in Figure 2.

Besides Aβ and Tau proteins, several other exosomal proteins have been shown to correlate with AD pathology. Among them, synaptic proteins like synaptophysin, synaptotagmins, synaptobrevin, synaptopodin, Ras-related protein Rab3A, Growth-associated protein-43 (GAP 43) and neurogranin were shown to be present in reduced quantity in the exosomal cargo in AD patients [156,157]. Altered levels of cathepsin D, lysosome-associated membrane protein 1 (LAMP1), ubiquitinylated proteins and heat-shock protein 70 were also observed in exosomes isolated from AD patients in comparison to controls [156]. In another study, reduced levels of low-density lipoprotein-receptor-related protein 6, heat-shock factor-1 and repressor element 1-silencing transcription factor, which are all involved in neuronal defenses against diverse stresses, were quantified in exosomes from AD patients with respect to healthy individuals [155]. Interestingly, also the level of different proteins somehow correlated to the antioxidant response was found altered in exosomes derived from AD brains. More specifically, increased levels of heat shock protein family A member 1A (HSPA1A), aminopeptidase Puromycin Sensitive (NPEPPS) and Prostaglandin F2 Receptor Inhibitor (PTGFRN) were found in the cerebrospinal fluid of AD patients and could represent useful markers to monitor the progression of the disease [158,159]. The increased expression of HSPA1A, which negatively regulated APP processing and Aβ production, has been suggested to be related to the rebound antioxidant response caused by ROS-mediated intracellular stress [160,161]. Instead, the involvement of NPEPPS in acting on neurotoxic Tau protein and protecting against Tau-induced neurodegeneration could represent a plausible explanation for the altered expression of NPEPPS [162].

### 4.2. Exosomal miRNAs and AD

Various studies have explored the potential of exosomal microRNAs as diagnostic biomarkers for AD, revealing distinctive expression patterns in plasma, serum and CSF. More specifically, the potential of circulating exosomal miRNA profiling via next-generation sequencing analysis has been suggested for AD [163]. However, inconsistencies in results from various studies are quite apparent. Nevertheless, several miRNAs related to APP processing, Aβ degradation, Aβ aggregation and Tau phosphorylation have shown significantly altered levels. The miRNAs like miR-15a-5p, miR-18b-5p, miR-20a-5p, miR-30e-5p, miR-93-5p, miR-101-3p, miR-106a-5p and miR-143-3p were increased while miR-15b-3p, miR-342-3p and miR-1306-5p were decreased in circulating peripheral exosomes [163,164]. In another study, altered expression levels of miR-23b-3p, miR-24-3p, miR-29b-3p, miR-125b-5p, miR-138-5p, miR-139-5p, miR-141-3p, miR-150-5p, miR-185-5p, miR-338-3p, miR-342-3p, miR-342-5p, miR-548at-5p, miR-3613-3p, miR-3916 and miR-4772-3p were detected in plasma derived exosomes from patients clinically diagnosed with AD dementia [165]. Similarly, multiple studies have highlighted the differential expression of exosomal miRNAs in CSF samples of AD patients. For instance, the altered expression of miR-29c, miR-136-3p, miR-16-2, miR-331-5p, miR-132-5p and miR-485-5p in AD patients compared to healthy controls was observed using a TaqMan miRNA array [166]. Accordingly, in a study based on young-onset AD patients, a decrease in miR-16-5p, miR-451a and miR-605-5p levels, and an enhanced amount of miR-125b-5p, was detected [167]. Moreover, another study involving comparative miRNA profiling analysis in whole CSF and in the CSF exosome-enriched fraction from AD patients and healthy controls identified 14 differentially expressed miRNA [168]. Likewise, differential expression profiles of several exosomal miRNAs can distinguish AD from other types of neurodegenerative diseases like Parkinson’s disease (PD) or dementia with Lewy bodies (DLB) [169,170]. Among the different miRNAs evaluated, miR-193b is one of the most characterized. Its levels were demonstrated to be downregulated both in CSF and plasma-derived exosomes in AD patients in comparison to control individuals, and a negative correlation between exosomal levels of miR-193b and Aβ1-42 levels emerged. Moreover, a bioinformatic analysis suggested the potential regulatory effect of miR-193b in APP expression [171]. Another miRNA, miR-342-3p, has also proven to be crucial for learning and memory function by modulating the APP and Tau processing [172,173].

Noteworthy for the use of miRNAs as biomarkers of AD, exosomes derived from both serum and plasma have shown promising association in AD patients along with correlation to the Mini-Mental State Examination score [163,165,174].

Experimental evidence indicates that astrocyte-derived small extracellular vesicles (sEVs) accumulate Aβ more in female transgenic mice. Additionally, these astrocyte-derived sEVs increase the toxicity of Aβ when absorbed by neurons [175,176]. The potential of sexual dimorphism in exosomal contents in AD pathology needs further in-depth analysis.

## 5. The Crosstalk between Exosomal miRNAs and Oxidative Stress in AD

The interdependent connections between oxidative stress and microRNAs reveal a complex network of interactions with implications for AD pathogenesis [177]. Oxidative stress has a significant effect on the complex regulation of miRNA expression, affecting several molecular components. In contrast, miRNAs exert their regulatory influence across a wide range of genes that are intimately engaged in the physiological response to oxidative stress.

Among these oxidative-stress-related miRNAs, miR-34a, known to suppress tumor progression, was suggested to be very relevant in AD pathology [177,178,179]. Different studies have also indicated a significantly increased level of miR-34a in the brain and peripheral blood mononuclear cells of AD patients and in animal models [180,181]. At the experimental level, overexpression of miR-34a in a mice model demonstrated cognitive deficits associated with altered APP processing, increased Aβ production and increased Tau phosphorylation [182,183]. Conversely, the miR-34a knockout-mouse model showed cognitive improvement via inhibition of the amyloidogenic processing of APP through attenuation of γ-secretase activity [184]. Further, the upregulation of miR-34a was also associated with the simultaneous downregulation of its target genes, impacting synaptic plasticity and decreasing ATP generation through the inhibition of oxidative phosphorylation and glycolysis [182]. The autophagic impairment, mitochondrial dysfunction and oxidative stress as a consequence of miR-34a expression observed in experimental models have been suggested as a potential route to influence the course of AD [185]. Another possible mechanism of miR-34a-mediated redox imbalance could be the downregulation of SIRT1 expression [186]. It is also interesting to note that exosomal release of miR-34a was demonstrated in cultured primary neurons overexpressing miR-34a, implying a neuron-to-neuron transfer of AD pathogenic mechanisms [182]. Despite the importance of miR-34a in AD pathogenesis in model systems as described here, an increased level of exosomal miR-34a in AD patients has not been established. Instead, a study demonstrated a decrease in plasma and CSF miR-34a in AD subjects [187]. The upregulation of another miRNA potentially related to the redox state, miR-125b-5p, emerged from a comparative analysis of CSF of AD patients and healthy volunteers [188]. Notably, increased miR-125b-5p levels could induce Tau hyperphosphorylation, neuronal apoptosis, oxidative stress and inflammation in experimental models which are key events associated with AD progression [188,189]. Furthermore, decreased miR-125b-5p levels effectively exhibited neuroprotective properties by lowering ROS levels [190], and a miR-125b-5p-mediated rescue from oxidative stress via downregulation of BACE1 has also been proposed [188,191,192]. The involvement of another miRNA, miR-141-3p, in mediating redox imbalance through the induction of mitochondrial dysfunction has been suggested; miR-141-3p has also been shown to mediate oxidative-stress-induced apoptosis in cardiac myocytes [192,193]. Thus, both miR-125b-5p and miR-141-3p may have some implications in AD pathogenesis especially in the context of oxidative stress. Surprisingly, however, exosomal levels of miR-141-3p and miR-125b-5p have been reported to be decreased in AD patients [165].

## 6. Exosomes as a Therapeutic Cargo against Oxidative Stress in AD

Several studies have emphasized the potential protective effects mediated by exosomes against oxidative conditions in different experimental models. For instance, exosomes derived from human cardiac resident mesenchymal progenitor cells were demonstrated to be enriched in superoxide dismutase, effectively reducing the levels of ROS and mitigating oxidative damage when administered in rat ventricular myocytes [194]. Similarly, exosomes obtained from primary fibroblasts of young human donors were able to ameliorate senescence-related tissue damage in fibroblasts from old individuals due to the intrinsic presence of glutathione-S-transferase, which was able to enhance the levels of reduced glutathione and minimize oxidative stress and lipid peroxidation [195].

As aforementioned, the participation of miRNAs enclosed inside exosomes is recognized as a key mechanism for their antioxidant properties. In this frame, miR-155-5p demonstrated distinct regulatory functions by suppressing genes associated with oxidative stress and reducing the expression of angiotensin-converting enzyme, therefore alleviating oxidative damage [196]. Further highlighting the influence of exosomal miRNAs on oxidative stress, exosomal miR-320a and miR-214 have important functions in decreasing the production of sirtuin 4 and suppressing the expression of calcium/calmodulin-dependent protein kinase II, respectively. These effects led to a reduction in the formation of ROS [197,198]. In another study, the beneficial effects of miR132-3p-enriched mesenchymal-stromal-cell-derived exosomes on oxidative stress, apoptosis, barrier disruption and cerebral injury were demonstrated in hypoxia/reoxygenated-injured endothelial cells and a mouse model of ischemic stroke. The miR132-3p-mediated protection was attributed to the ability of the miRNA to activate the PI3K/Akt/eNOS pathway [199].

Importantly, the protective effects of exosomes against oxidative stress were also observed in AD-related models. In neuron primary cultures from an AD mouse model, exosomes produced from human amniotic fluid stem cells (AFSC-exos) were shown to play a protective function by raising the expression of antioxidant enzymes. This rise led to a decrease in ROS levels, addressing a major element of oxidative stress in AD. The activation of the PI3K/Akt signaling pathway along with the suppression of NOX4 (gene encoding NADPH oxidase 4) was then identified as a contributing component to the antioxidant activity of AFSC-exos [200]. Furthermore, exosomes obtained from mesenchymal stem cells showed the capability to protect hippocampal neurons from oxidative stress and synapse damage induced by Aβ oligomers. Protection was described as depending on the presence of catalase inside exosomes [201].

## 7. Conclusions

The challenges in diagnosing AD underscore the need for novel early biomarkers. The cascade of events triggered by oxidative stress is a fundamental aspect of the development of AD. Concurrently, the crucial role of oxidative stress in altering the biogenesis, content, and quality of exosomes has obscured the focus on BDEs as diagnostic indicators of AD. Furthermore, elucidating the interplay between exosomal miRNAs and oxidative stress may provide new clues to our understanding of the complex molecular mechanisms of AD. The idea of using exosomes as indicators to address oxidative stress in AD shows promise as a possible therapeutic approach, opening up possibilities for future therapies. However, it is worth mentioning that this topic is an emerging one, and there are multiple lacunae in our understanding of the significance of exosomes vis a vis the oxidative stress mechanism of AD pathogenesis.

## Figures and Tables

**Figure 1 antioxidants-13-00316-f001:**
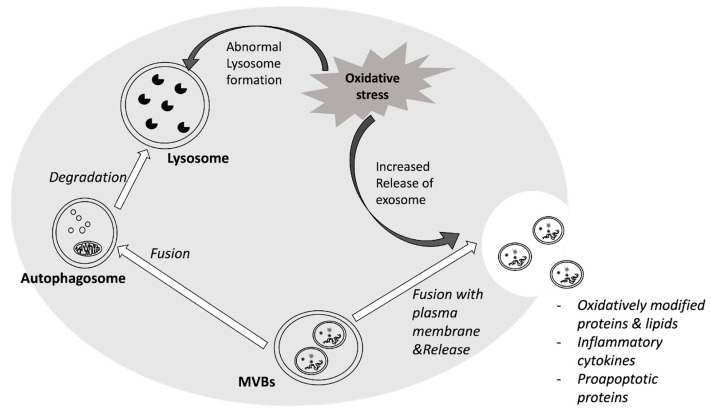
Oxidative stress impact on exosomes. ROS imbalance modifies the process of biogenesis and the release of exosomes. Prolonged or severe oxidative stress can also alter their composition.

**Figure 2 antioxidants-13-00316-f002:**
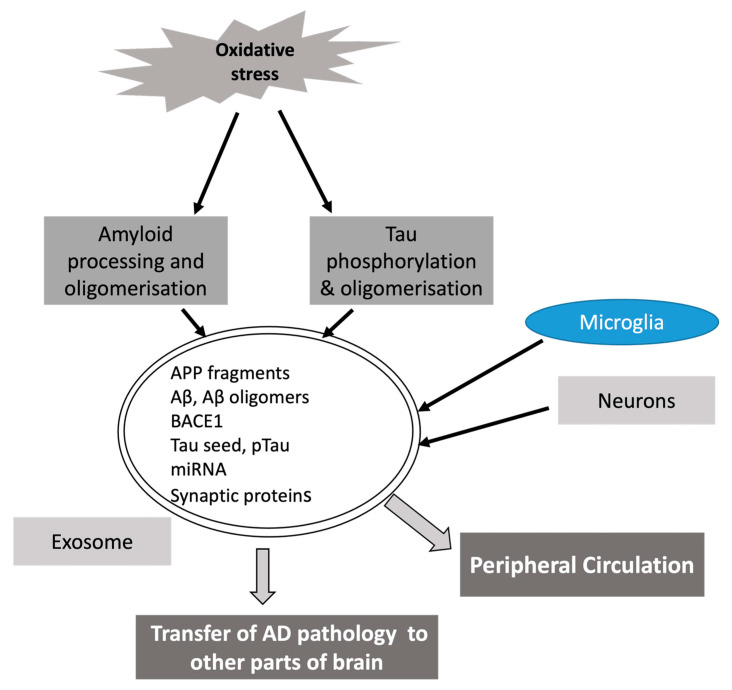
Role of exosomes in AD. Oxidative stress contributes to promoting Aβ and Tau pathology, which can then spread throughout the brain through the participation of exosomes derived from both neurons and microglia.

## Data Availability

Not applicable.

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
