# Peer review of "Exosomal Dynamics and Brain Redox Imbalance: Implications in Alzheimer’s Disease Pathology and Diagnosis"

_antioxidants, 2024, doi:10.3390/antiox13030316_

Round 1

Reviewer 1 Report

Bir et al. presented exosomal dynamics and brain redox imbalance, implicating pathology and diagnosis of AD. This is an interesting review paper, indicating 

 the potential of exosomes as a novel tool in AD diagnosis. Furthermore, the authors illuminate the interplay between exosomal miRNAs and oxidative stress, providing new clues to our understanding of the complex molecular mechanisms of AD. The paper is well written. There are, however, some issues to be addressed to further improve the manuscript.

1.     Exosomes are able to cross the blood brain barrier (BBB) from the brain to the bloodstream as well as from the blood to the central nervous system. The transport mechanism through BBB taken by exosomes should be discussed.

2.     The relationship of peripheral exosomes and preclinical AD prior to MCI would be intriguing and should be discussed.

3.     Given that sex and aging differentially modulate brain networks, sex-dependent differences in exosome levels and content in the brain in AD patients should be reviewed.

Author Response

We very much appreciate the general positive comments on our manuscript and also would like to respond to specific issues raised by the reviewer.

Q1. Exosomes are able to cross the blood brain barrier (BBB) from the brain to the bloodstream as well as from the blood to the central nervous system. The transport mechanism through BBB taken by exosomes should be discussed.

R1. We thank the reviewer for this pertinent comment. We have briefly discussed it in the revised version with references under 3.1.

Q2. The relationship of peripheral exosomes and preclinical AD prior to MCI would be intriguing and should be discussed.

R2. We thank the reviewer again for an interesting suggestion. We have briefly discussed this aspect in the revised version (4.1).

Q3. Given that sex and aging differentially modulate brain networks, sex-dependent differences in exosome levels and content in the brain in AD patients should be reviewed.

R3. We have complied with this suggestion in the revised version (4.2) with additional references.

Reviewer 2 Report

The authors discuss exosome dynamics and the imbalance of brain redox, particularly in relation to Alzheimer's disease (AD). The molecular phenomena of Abeta and Tau protein as models of AD aggregation are outlined, and the relationship between metal ion homeostasis and oxidative stress and the presence of miRNAs in exosomes are well described. In addition to the overall coverage of the topics, the explanations are clear and concise. Therefore, the review is well-written and generally free of major issues requiring revision. 

In my opinion, Section 3 and 4 could be combined, and it might be beneficial to use subsections for better organization.

Additionally, there are some typos that need careful confirmations. 

Author Response

We are thankful to the reviewer for the positive comments on our manuscript. We also would like to respond to the specific comments.

Q1. In my opinion, Section 3 and 4 could be combined, and it might be beneficial to use subsections for better organization.

Additionally, there are some typos that need careful confirmations.

R1. We fully agree with the reviewer that the manuscript will look better organized with subsections, and we have complied with this in the revised version (sections 2, 3 and 4 in particular). However, we feel that it is better to keep sections 3 and 4 separated in the manuscript. We have taken care of the typos.

Reviewer 3 Report

Bir and colleagues have reviewed the interconnection of exosomes and redox imbalance in Alzheimer’s disease. The review is of interest and thorough work highlighting the different aspects related to oxidative stress and exosomes. There are only some remarks:

Oxidative stress is characteristic for not only AD, but other neurodegenerative diseases such as Parkinson’s disease (PD) as well. The authors describe that the content of brain-derived exosomes can be used to distinguish AD and control samples. Are there studies which compare exosomes from AD patients with those from other diseases? Are the alterations specific for AD? For example, alteration of synaptic proteins may be characteristic for PD as well. I suggest that authors should mention these aspects.

It is not clear whether the authors discuss only brain-derived exosomes or alterations in other exosomes as well. I would suggest to include brain-derived exosomes in the title, if that is the case.

lines 411-414: Were these observations found in PD or AD patients?   

Author Response

We thank the reviewer for an overall positive evaluation. We also answer the specific questions raised by the reviewer.

Q1. The authors describe that the content of brain-derived exosomes can be used to distinguish AD and control samples. Are there studies which compare exosomes from AD patients with those from other diseases? Are the alterations specific for AD? For example, alteration of synaptic proteins may be characteristic for PD as well. I suggest that authors should mention these aspects.

R1. In most studies related to exosomal contents in AD, comparisons have been made between controls and AD subjects. However, some studies have examined and compared the exosomal contents (both synaptic proteins and miRNAs) from AD with other neurodegenerative diseases such as frontotemporal dementia, dementia with Lewey body, PD etc. This has been mentioned in the revised version (subsections 4.1 and 4.2).

Concerning the specificity of alterations in exosomal contents in AD, it is difficult to comment in the absence of a large number of validated studies with AD and related dementias, especially in the case of miRNAs. However, we presume that certain alterations could be specific for AD (Aβ, Tau or pTau, APP fragments or other proteins directly involved in AD pathogenesis); but we have mentioned in the “Conclusions” that this is an emerging science and more explorations are necessary to fill the knowledge gaps.

Q2. It is not clear whether the authors discuss only brain-derived exosomes or alterations in other exosomes as well. I would suggest to include brain-derived exosomes in the title, if that is the case.

R2. We thank the reviewer for a pertinent comment. In large parts of the text, we discussed about exosomes in general. In the context of AD mostly brain-derived exosomes or neuronal-derived exosomes have been discussed and these have been mentioned in multiple places in the text. However, many original studies have used circulating exosomes without enriching the fraction with brain-derived exosomes. Moreover, multiple preparative methods have been used by the researchers to isolate exosomes resulting in variability in the study outcomes. We feel that under such circumstances it is better to use the general term exosomes in the text and the title.

Q3. lines 411-414: Were these observations found in PD or AD patients? 

R4. The observations were in AD patients; the mistake is corrected in the revised version.

Reviewer 4 Report

In this review article the authors explore exosomal dynamics and brain redox imbalance their implications in Alzheimer’s disease pathology and diagnosis. This is an important topic and deserved attention with the potential of making significant progress in Alzheimer’s disease pathophysiology. The review in its current stage needs substantial revision as too many concepts are assumed without introduction or defining them in the text. This makes the review difficult to follow see detail below. Also, the link between exosome and brain redox imbalance and their association with Alzheimer’s disease is missing or not clearly illustrated in the review. Furthermore, how is exosome used to mitigate against oxidative damage in Alzheimer’s disease progression and diagnosis and potential pitfalls of this techniques.  

1.       It is important to define key major concepts from the beginning of the review so that it becomes easy to follow. What is exosome, how are they involved in health and disease stage? What is redox imbalance, redox-active and inactive, oxidative stress and how are they associated with Alzheimer’s disease.

2.       The authors mentioned that fascinating observation on the interactions of oxidative stress and exosomes are being reported without mentioning the observation.

3.       The authors mentioned that extensively studies about oxidative stress and A beta without providing and or describing these findings but only gave one reference. How is this extensive?

4.       How is the binding of metals and A beta facilitate redox-cycling activities?

5.       The authors mentioned that the hypothesis is supported by scattered experimental evidence showing an antioxidant and protective role of A beta. What is the source no reference provided?

6.       Figure 2 legend additional description of the events leading to AD pathology.

In this review article the authors explore exosomal dynamics and brain redox imbalance their implications in Alzheimer’s disease pathology and diagnosis. This is an important topic and deserved attention with the potential of making significant progress in Alzheimer’s disease pathophysiology. The review in its current stage needs substantial revision as too many concepts are assumed without introduction or defining them in the text. This makes the review difficult to follow see detail below. Also, the link between exosome and brain redox imbalance and their association with Alzheimer’s disease is missing or not clearly illustrated in the review. Furthermore, how is exosome used to mitigate against oxidative damage in Alzheimer’s disease progression and diagnosis and potential pitfalls of this techniques.  

1.       It is important to define key major concepts from the beginning of the review so that it becomes easy to follow. What is exosome, how are they involved in health and disease stage? What is redox imbalance, redox-active and inactive, oxidative stress and how are they associated with Alzheimer’s disease.

2.       The authors mentioned that fascinating observation on the interactions of oxidative stress and exosomes are being reported without mentioning the observation.

3.       The authors mentioned that extensively studies about oxidative stress and A beta without providing and or describing these findings but only gave one reference. How is this extensive?

4.       How is the binding of metals and A beta facilitate redox-cycling activities?

5.       The authors mentioned that the hypothesis is supported by scattered experimental evidence showing an antioxidant and protective role of A beta. What is the source no reference provided?

6.       Figure 2 legend additional description of the events leading to AD pathology.

Author Response

We are grateful to the reviewer for taking a serious interest in our manuscript and providing very constructive criticisms. We would like to respond to the specific comments mentioning the modifications done in the revised version.

Q1.  It is important to define key major concepts from the beginning of the review so that it becomes easy to follow. What is exosome, how are they involved in health and disease stage? What is redox imbalance, redox-active and inactive, oxidative stress and how are they associated with Alzheimer’s disease.

R1. We thank the reviewer for the very pertinent and useful suggestion. The key concepts of redox imbalance, oxidative stress, redox-active and inactive metals have been developed in the revised version with substantial additions in the text with multiple references (section 2). The relationship of oxidative stress, metal dyshomeostasis, Aβ-metal interactions and others with AD pathogenesis have been clearly stated in the revised version (section 2 and subsections  2.5, 2.6).  Multiple references have been added. The biogenesis of exosomes, their transport and other related information have been provided (section 3).

Q2. The authors mentioned that fascinating observation on the interactions of oxidative stress and exosomes are being reported without mentioning the observation.

R2. Corrected in the revised version.  

Q3. The authors mentioned that extensively studies about oxidative stress and A beta without providing and or describing these findings but only gave one reference. How is this extensive?

R3. We have added now 6 references (6,50, 51,52,53,54) to support this point in the revised version in subsection 2.2. We thank the reviewer for a careful scrutiny.

Q4. How is the binding of metals and A beta facilitate redox-cycling activities?

R4. We have explained this point in the revised version with references (subsection 2.5).

Q5. The authors mentioned that the hypothesis is supported by scattered experimental evidence showing an antioxidant and protective role of A beta. What is the source no reference provided?

R5. We thank the reviewer for pointing out an error. We have added an important reference (58) in this context. Moreover, the references 59 and 60 also highlight this point (subsection 2.2). The reference 60 is a review that cites multiple experimental studies indicating the antioxidative role of Aβ.

Q6. Figure 2 legend additional description of the events leading to AD pathology.

R6. The additional description has been provided in Fig.2 legend.

Round 2

Reviewer 1 Report

The authors put an effort in revising their manuscript and addressing issues raised previously. The paper was improved.

Nothing particular.

Reviewer 4 Report

The authors have addressed all my major concerns on their manuscript.

The authors have addressed all my major concerns on their manuscript.